

# CrossAlignNet: a self-supervised feature learning framework for 3D point cloud understanding

Fei Wang[1], Xingzhen Dong[1], Jia Wu[1], Weishi Zhang[1] and Tuo Zhou[2,3]

[1] School of Information Science and Technology, Dalian Martime University, Dalian, China
[2] School of Information and Electronic Engineering, Shandong Technology and Business University, Yantai, China
[3] Research Institute of Information Fusion, Naval Aviation University, Yantai, China

## ABSTRACT

We propose a self-supervised point cloud representation learning framework CrossAlignNet based on cross-modal mask alignment strategy, to solve the problems of imbalance between global semantic and local geometric feature learning, as well as cross-modal information asymmetry in existing methods. A geometrically consistent mask region is established between the point cloud patches and the corresponding image patches through a synchronized mask alignment strategy to ensure cross-modal information symmetry. A dual-task learning framework is designed: the global semantic alignment task enhances the cross-modal semantic consistency through contrastive learning, and the local mask reconstruction task fuses the image cues using the cross-attention mechanism to recover the local geometric structure of the masked point cloud. In addition, the ShapeNet3D-CMA dataset is constructed to provide accurate point cloud-image spatial mapping relations to support cross-modal learning. Our framework shows superior or comparative results against existing methods on three point cloud understanding tasks including object classification, few-shot classification, and part segmentation.

## INTRODUCTION

As a fundamental 3D spatial data representation, point clouds enable precise modeling of geometric structures through direct coordinate measurements. This data modality has gained ubiquitous applications across computer vision and unmanned systems such as autonomous driving, augmented reality and robotics. Nevertheless, supervised deep learning paradigms exhibit inherent limitations in scalability for point cloud comprehension tasks, primarily constrained by their heavy reliance on extensive labeled datasets. Such dependency inevitably creates dual operational challenges: prohibitive annotation costs and compromised efficiency when deployed in complex real-world environments.

To address these limitations, point cloud representation learning has emerged as a transformative paradigm. Through carefully designed self-supervised pretext tasks, this approach extracts discriminative, robust, and interpretable features from raw point cloud

Corresponding author
Weishi Zhang, teesiv@dlmu.edu.cn

data characterized by disorder, sparsity, and non-structural organization. The learned representations effectively support various downstream applications including but not limited to object classification, detection, semantic segmentation, and generative modeling.

Current self-supervised representation learning approaches (*Zeng et al., 2024*; *Xiao et al., 2023*) for point clouds predominantly bifurcate into two technical paradigms: global representation learning and local representation learning. The global paradigm primarily emphasizes cross-modal semantic consistency, operating under the foundational premise that multiple augmented views of identical objects should exhibit feature space congruence, whereas distinct objects maintain feature divergence. This framework typically employs contrastive learning frameworks to derive discriminative global semantics. A representative implementation, Crosspoint (*Afham et al., 2022*), establishes 3D-2D cross-modal correspondence through invariant space alignment between point clouds and their rendered 2D counterparts, simultaneously enforcing transformation invariance within the point cloud domain. Building upon this, CrossNet (*Wu et al., 2024*) innovatively decomposes rendered images into RGB and grayscale components, enabling separate extraction of chromatic features and geometric descriptors, which are subsequently aligned with their 3D representations. However, global representation learning methods demonstrate limited efficacy in capturing fine-grained geometric patterns due to its global feature aggregation mechanism. Consequently, while achieving superior performance in object classification tasks, its applicability diminishes in scenarios requiring granular structural understanding, particularly in point cloud segmentation applications.

Local representation learning frameworks predominantly employ masked reconstruction mechanisms to preserve geometric fidelity at fine scales. Typical implementations like Point-Bidirectional Encoder Representations from Transformers (BERT) (*Yu et al., 2022*) and Point-Masked Autoencoder (MAE) (*Pang et al., 2022*) exemplify this strategy through distinct masking approaches. Point-BERT partitions point clouds into localized patches, embeds them into discrete token sequences *via* vector quantization, and subsequently masks random token subsets for prediction tasks. This approach emphasizes learning contextual relationships between visible and occluded regions. In contrast, Point-MAE adapts the masked autoencoder architecture from visual domains, selectively obscuring substantial point subsets (typically 60–80%) while requiring coordinate reconstruction of occluded regions. Notably, both methodologies share a unified objective: to derive discriminative local geometric representations through self-supervised reconstruction. However, their independent masking protocols introduce critical limitations in cross-modal learning scenarios. Specifically, modality-specific masking patterns disrupt inherent cross-modal correlations, resulting in asymmetric feature preservation across modalities that degrades representation alignment.

To overcome these limitations, we propose CrossAlignNet, a dual-stream cross-modal framework that synergistically learns global semantics and local geometric representations of point clouds. Our architecture integrates two complementary pretraining objectives: global semantic alignment (GSA) targeting cross-modal consistency, and local mask reconstruction (LMR) focusing on geometric detail restoration. The core innovation lies in

our synchronized masking strategy for cross-modal inputs. For paired point clouds and rendered images, we implement geometrically aligned masking that maintains strict cross-modal correspondence. This strategy ensures equivalent information retention across modalities, effectively mitigating feature asymmetry caused by independent masking schemes. The GSA module processes visible patches from both modalities through contrastive learning in a shared embedding space, maximizing mutual information between matched pairs. The LMR component employs coordinate regression to predict masked 3D structures, using cross-attention mechanisms to fuse visual cues from corresponding image patches. In addition, it is difficult to obtain the correspondence between points and pixels in existing cross-modal datasets. For this reason we construct a cross-modal representation learning dataset based on the ShapeNet V2 dataset to support the learning of global semantic and local geometric representations.

Our main contributions are as follows:

- We propose a dual-stream cross-modal framework to synergistically learn global semantics and local geometric representations for point cloud understanding tasks.
- A geometrically constrained masking strategy is proposed to establish bi-directional correspondence between 3D point clusters and 2D image patches, effectively resolving modality-specific information asymmetry.
- We introduce ShapeNet3D-CMA, a large-scale dataset for representation learning featuring: precise point cloud-to-pixel spatial mappings, and multi-view photorealistic renderings with calibrated camera parameters.
- Extensive experiments demonstrate superior performance across object classification (84.5% accuracy on ScanObjectNN), Few-shot classification (91.8% with 10-shot samples), and part segmentation (84.2% mIoU on ShapeNetPart).

## RELATED WORK

Recent advances in masked self-supervised learning for point cloud representation demonstrate progressive refinement in geometric relationship modeling through innovative masking strategies (*Wang et al., 2021*; *Yu et al., 2022*; *Li et al., 2022*; *Pang et al., 2022*; *Zhang et al., 2022a*, *2023*; *Chen et al., 2023*; *Tang et al., 2024*; *Lin et al., 2024*; *Zeng et al., 2024*; *Xiao et al., 2023*). The seminal work Occlusion Completion (OcCo) (*Wang et al., 2021*) introduced viewpoint-aware occlusion simulation, employing encoder-decoder architectures for spatial completion tasks. Subsequent innovations like Mask-Point (*Li et al., 2022*) established discriminative learning paradigms by partitioning clouds into masked/unmasked regions and designing binary classification tasks to authenticate reconstructed points.

The emergence of Transformer-based frameworks marked a paradigm shift. Point-MAE (*Pang et al., 2022*) adapted masked autoencoding principles to point clouds through coordinate reconstruction objectives. Building upon this, Point-M2AE (*Zhang et al., 2022a*) developed hierarchical multi-scale transformers to capture structural dependencies across spatial resolutions. Concurrently, Image-to-Point Masked

Autoencoders (I2P-MAE) (*Zhang et al., 2023*) enhanced 3D representation fidelity by incorporating distilled knowledge from 2D vision transformers. Recent developments focus on masking strategy optimization. Point-LGMask (*Tang et al., 2024*) strengthens local-global interactions through adaptive multi-ratio masking, while PatchMixing Masked Autoencoders (PM-MAE) (*Lin et al., 2024*) introduces momentum contrast with compound masking protocols. These approaches systematically address the dual challenges of geometric detail preservation and contextual relationship modeling.

Contrastive learning frameworks (*Chhipa et al., 2022*; *Xie et al., 2020*; *Long et al., 2023*; *Yin et al., 2022*; *Afham et al., 2022*; *Wu et al., 2024*; *Liu et al., 2024*; *Zeng et al., 2024*; *Xiao et al., 2023*) have significantly advanced global semantic abstraction in point cloud representation through innovative pretext task designs. DepthContrast (*Chhipa et al., 2022*) pioneers geometric invariance learning by generating positive pairs *via* depth map augmentation and modality conversion, followed by contrastive loss formulation in global feature space. Subsequent works refine the contrastive paradigm through spatial correspondence constraints. PointContrast (*Xie et al., 2020*) establishes multi-view consistency by maximizing feature similarity of overlapping regions while repelling non-overlapping areas. This approach effectively aligns cross-view geometric contexts through region-aware contrastive optimization. Further advancements integrate hierarchical feature learning mechanisms. PointClustering (*Long et al., 2023*) introduces transfer variance regularization during multi-perspective feature aggregation, combining point-wise and instance-level transformations to enhance cluster discrimination. In parallel, ProposalContrast (*Yin et al., 2022*) shifts the contrastive granularity to region-level proposals, enforcing semantic consistency across augmented 3D observation samples through proposal matching objectives.

Cross-modal self-supervised learning for point cloud representation (*Xu et al., 2022*; *Afham et al., 2022*; *Zhang et al., 2023*; *Chen et al., 2023*; *Zhou et al., 2024*; *Wu et al., 2024*; *Liu et al., 2024*; *Zeng et al., 2024*; *Xiao et al., 2023*) enhances feature discriminability through synergistic fusion of multimodal data. Current methodologies primarily exploit visual and linguistic modalities for complementary learning. Pioneering work by *Jing, Zhang & Tian (2021)* established cross-modal consistency discrimination frameworks through point cloud-image feature alignment. Following this, Image2Point (*Xu et al., 2022*) achieves parameter-level knowledge transfer from 2D pre-trained models *via* structural mapping. CrossPoint (*Afham et al., 2022*) advances differentiable rendering alignment to maximize geometric-semantic consistency between 3D points and 2D projections. Recent innovations integrate reconstruction objectives with visual cues. I2P-MAE (*Zhang et al., 2023*) jointly optimizes depth prediction and semantic reconstruction across modalities. Point Cloud and Image Interactive Masked Autoencoders (PiMAE) (*Chen et al., 2023*) introduces complementary masking with dual reconstruction losses for collaborative optimization. CrossNet (*Wu et al., 2024*) and Inter-MAE (*Liu et al., 2024*) further disentangle cross-modal features through contrastive subspace learning, while PointCMC (*Zhou et al., 2024*) establishes hierarchical correspondence *via* multi-scale graph matching.

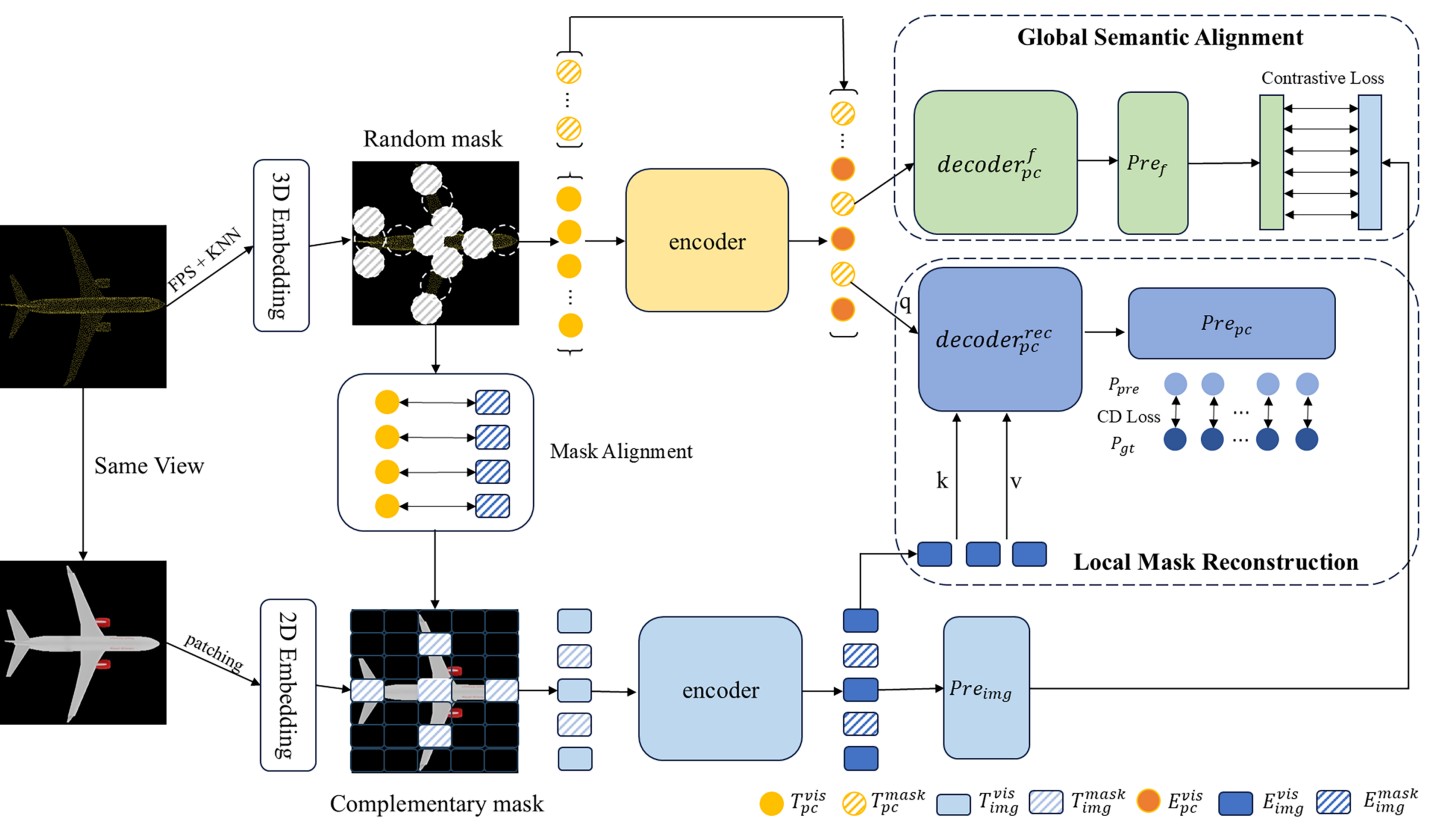

**Figure 1 Structure of our CrossAlignNet for global semantic and local geometry feature learning of point clouds.**

Semantic-aware methods (*Zhang et al., 2022b*; *Zhu et al., 2023*; *Huang et al., 2024*; *Xue et al., 2023*, *2024*) bridge 3D geometry with linguistic semantics. PointCLIP (*Zhang et al., 2022a*) pioneers zero-shot 3D recognition by projecting point clouds to CLIP's visual-textual space, albeit constrained by sparse depth representations. PointCLIP V2 (*Zhu et al., 2023*) overcomes these limitations through learnable projection modules and GPT-3 optimized prompt engineering.

Inspired by PiMAE (*Chen et al., 2023*) and Inter-MAE (*Liu et al., 2024*), we introduce a masked alignment strategy into a dual-task self-supervised learning framework, addressing key challenges in cross-modal Point cloud representation learning: asymmetrical modality information retention and imbalanced learning of global semantics and local geometric features.

## METHOD

### Overview

The structure of our CrossAlignNet framework is shown in Fig. 1. It takes the same viewpoint-aligned point cloud and its rendered image as input. It constructs two branches of image feature extraction and point cloud feature extraction. To achieve self-supervised feature learning, we establish a global semantic contrastive learning task and a local geometric reconstruction task, respectively. Firstly, the farthest point sampling algorithm

(FPS) and the K-nearest neighbor algorithm (KNN) are used to sample the point cloud into point cloud patches. Image patches are generated through 2D convolution operations. Then, the mask alignment strategy is used to synchronize the masked point cloud patches and the image patches. Visible point cloud patches and image patches are fed into separate encoders to extract the modality-specific features.

In the global semantic alignment module, the point cloud features extracted by the encoder are fed to the decoder to extract the global semantic information. It is mapped to the cross-modal global semantic space through the prediction header module, thus constituting a contrastive learning task with the image global semantic information. In the local mask reconstruction module, the cross attention mechanism is introduced. Image token features constitute the key and value of the attention module; and point cloud token features constitute the query. The point cloud prediction head reconstructs the coordinates of the masked point cloud based on the query-completion mechanism. The Chamfer Distance is introduced as the loss function of local mask reconstruction task.

## Mask alignment strategy

Current cross-modal masking approaches exhibit modality isolation in occlusion pattern generation, creating critical representation learning bottlenecks (*Liu et al., 2024*). Specifically, independent masking operations between point clouds and images induce two inherent flaws: (1) spatial mismatch where masked 3D regions remain visible in 2D projections, and (2) asymmetric information retention that disrupts cross-modal feature correlation. These limitations fundamentally constrain joint global-local feature learning across modalities.

To address this problem, we propose a mask alignment strategy to establish the cross-modal data correspondence between the point cloud and the image to enhance the effectiveness of feature learning. The main idea is to first chunk and embed the input point cloud and image. Then the visibility of the corresponding image blocks is constrained synchronously when masking the point cloud regions by establishing the point cloud-image block-level correspondence as shown in Fig. 2. Detailed approaches are discussed as follows.

**Patches Generation**. For the input point cloud data $X \in \mathbb{R}^{n \times 3}$, we adopt the FPS to select key points and construct local neighborhoods *via* the KNN. This will generate m point cloud patches $PP \in \mathbb{R}^{m \times k \times 3}$. These patches are then embedded into $T_{pc}$ using a mini-PointNet (*Qi et al., 2017a*).

For image data $I \in \mathbb{R}^{c \times h \times w}$, we uniformly partition it into equally-sized image patches, which undergo 2D convolutional operations and are flattened into embedded token patches $T_{img}$.

**Masking**. For the point cloud patches, we use random masking process to get the visible marker patches $T_{pc}^{vis}$ with masked markers $T_{pc}^{mask}$. For the image patches, we use aligned masking process. First, we take the center point of the masked point cloud patches

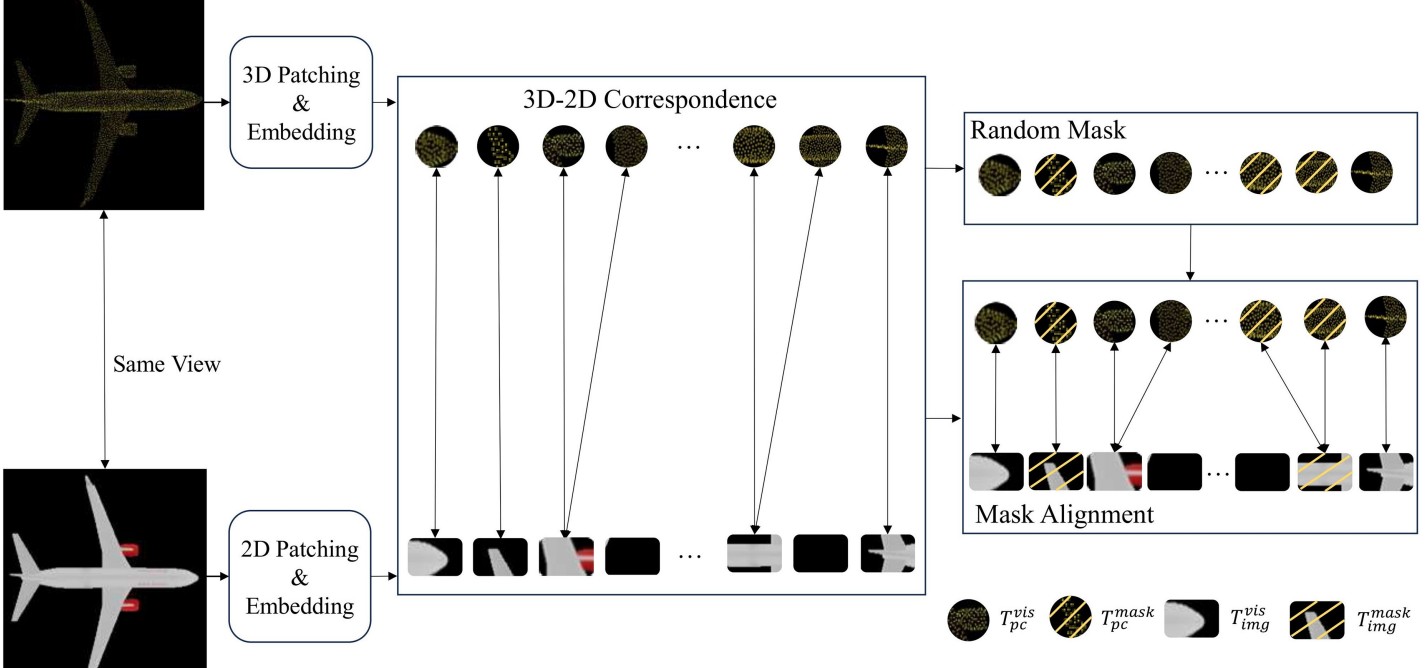

**Figure 2 Illustration of our mask alignment strategy between point cloud patches and image patches.**

through the extrinsic parameters matrix $R_t$ and the intrinsic parameters matrix $K$ to get its pixel coordinates $(u, v)$ projected in the image. The specific formula can be described as follows:

$$\begin{bmatrix} u \\ v \\ z \end{bmatrix} = proj\,(n) = K \cdot R_t \cdot \begin{bmatrix} x \\ y \\ z \\ 1 \end{bmatrix}. \qquad (1)$$

After obtaining the pixel coordinates $(u, v)$ corresponding to the center of each point cloud patch, the mask correspondence between the image and point cloud is established based on these coordinates. Assuming the image size is H×W and the image is divided into patches of size S×S (each assigned an index), the related image patch index $I_p$ can be calculated by the following formula:

$$I_p = \left\lfloor \frac{v}{S} \right\rfloor \times \left\lfloor \frac{W}{S} \right\rfloor + \left\lfloor \frac{u}{S} \right\rfloor. \qquad (2)$$

If the center of a point cloud patch is masked, the corresponding image patch containing the pixel coordinates $(u, v)$ is also masked. If a same image patch corresponds to a masked point cloud patch and a visible point cloud patch, then both the image patch and the two point cloud patches will be masked. After this, the visible tokens of the point

cloud $T_{pc}^{vis}$ correspond to the visible tokens of the image $T_{img}^{vis}$, while the masked tokens of the point cloud $T_{pc}^{mask}$ correspond to the masked tokens of the image $T_{img}^{mask}$.

## Encoders

In order to facilitate the comparative analysis, Transformer-based point cloud and image encoders are chosen as feature extraction in this article. However, the proposed method is not limited to this one encoder structure, and the generalization of the learned features may be further enhanced if a complex encoder structure is used.

For simplicity and generality, both the point cloud encoder $Encoder_{pc}$ and the image encoder $Encoder_{img}$ adopt a standard Transformer-based architecture, consistent with Point-MAE (*Pang et al., 2022*). The point cloud encoder processes only the visible token patches $T_{pc}^{vis}$ and outputs the feature $Encoder_{pc}$, while the image encoder processes both the visible token patches $T_{img}^{vis}$ and the masked token patches $E_{img}^{mask}$, generating the corresponding visible feature $E_{img}^{vis}$ and masked feature $E_{img}^{mask}$. This process can be formally defined as:

$$E_{pc}^{vis} = Encoder_{pc}(T_{pc}^{vis}) \tag{3}$$
$$E_{img}^{vis}, E_{img}^{mask} = Encoder_{img}(Concat(T_{img}^{vis}, T_{img}^{mask})). \tag{4}$$

## Self-supervised tasks

### Global semantic alignment

In the global semantic alignment task, a decoder $Decoder_{pc}^{f}$ is used to extract semantic information from the point cloud features obtained by the encoder. Then the global semantic features of the visible point cloud patches are generated and mapped to the semantic space shared with the image branch by the feature prediction head $F_{pc}$. For the image branch, we only use the image projection head $Pro_{img}$ to extract the global semantic feature of the image. In this case, the decoder $Decoder_{pc}^{f}$ for the feature prediction task is structured similarly to the encoder, but with fewer Tranformer blocks. The feature prediction header $F_{pc}$ has the same structure as the image projection header $Pro_{img}$, both consisting of two fully connected layers, and we use batch normalization and nonlinear activation only after the first layer.

Specifically, the visible point cloud tokens $E_{pc}^{vis}$ and the learnable point cloud mask tokens $E_{pc}^{mask}$ are concatenated and fed into $Decoder_{pc}^{f}$. The output is then passed through the feature prediction head to obtain the predicted feature vector $F_{pc}$. Simultaneously, the image tokens are mapped to the image feature vector $F_{img}$ *via* the projection head $Pro_{img}$. This process can be formally expressed as:

$$F_{pre} = Pre_f(Decoder_{pc}^{f}(E_{pc}^{vis}, E_{pc}^{mask})) \tag{5}$$
$$F_{img} = Pro_{img}(concat(E_{img}^{vis}, E_{img}^{mask})). \tag{6}$$

According to the idea of contrastive learning, point cloud and image data under the same viewpoint can be treated as positive sample pairs, and point cloud and image data under different viewpoints can be treated as negative sample pairs. For each sample pair

$(F_{pc}^i, F_{img}^i)$, a contrastive learning task can be set up, and its loss function is calculated as follows:

$$l(i, F_{pre}, F_{img}) = -\log \frac{e^{\langle F_{pre}^i, F_{img}^i \rangle / \tau}}{\sum_{k \neq i}^N e^{\langle F_{pre}^i, F_{img}^K \rangle / \tau} + \sum_i^N e^{\langle F_{pre}^i, F_{img}^K \rangle / \tau}} \quad (7)$$

$$Loss_f = \frac{1}{2N} \sum_{i=1}^N l(i, F_{pre}, F_{img}) + l(i, F_{img}, F_{pre}). \quad (8)$$

### Local mask reconstruction

This task achieves fine local geometric reconstruction through a cross-modal feature fusion mechanism. The decoder $Decoder_{pc}^{rec}$ is composed of cross-attention and self-attention layers, while the point cloud prediction head $Pre_{pc}$ consists of a simple fully connected layer.

Specifically, the visible point cloud tokens $E_{pc}^{vis}$, the masked point cloud tokens $E_{pc}^{mask}$, and the visible image tokens $E_{img}^{vis}$ are jointly fed into $Decoder_{pc}^{rec}$. After processing by the decoder, the intermediate feature $D_{pre}$ is obtained. This feature is then passed through the prediction head and undergoes dimension reshaping to generate the reconstructed point cloud patches $P_{pre}$. The process can be formally expressed as:

$$D_{pre} = Decoder_{pc}^{rec}(concat(E_{pc}^{vis}, E_{pc}^{mask}), E_{img}^{vis}) \quad (9)$$

$$P_{pre} = Reshape(Pre_{pc}(D_{pre})). \quad (10)$$

Reshape($\cdot$) operation reconstructs the vector into a $K3$ dimensional coordinate matrix ($K$ denotes the number of points per patch). The reconstruction objective minimizes coordinate errors in masked regions through an enhanced Chamfer Distance metric between predicted patches $P_{pre} \in \mathbb{R}^{K \times 3}$ and ground truth patches $P_{gt} \in \mathbb{R}^{K \times 3}$:

$$Loss_{pc} = \frac{1}{|P_{pre}|} \sum_{x \in P_{pre}} \min_{y \in P_{gt}} \|x - y\|_2^2 + \frac{1}{|P_{gt}|} \sum_{x \in P_{gt}} \min_{y \in P_{pre}} \|x - y\|_2^2. \quad (11)$$

### Training

During pre-training, the selected point cloud samples are downsampled to 2,048 points using the FPS algorithm and partitioned into 64 point cloud patches, each containing 32 points. The corresponding view-aligned images are resized to $224 \times 224$ as input. No data augmentation is applied except normalization to both images and point clouds. The pre-training employs the AdamW optimizer (*Loshchilov & Hutter, 2017*) for 300 epochs with a cosine decay learning rate schedule (*Loshchilov & Hutter, 2016*), initial learning rate of 0.001, weight decay of 0.05, and batch size of 128. A masking ratio of 60% is adopted during pre-training. After pre-training, only the point cloud Transformer encoder is transferred as initial weights for downstream tasks. The total optimization objective comprises two components: the feature prediction task loss and the mask completion task loss. The overall loss is defined as:

$$Loss_{total} = Loss_{pc} + \lambda Loss_f. \quad (12)$$

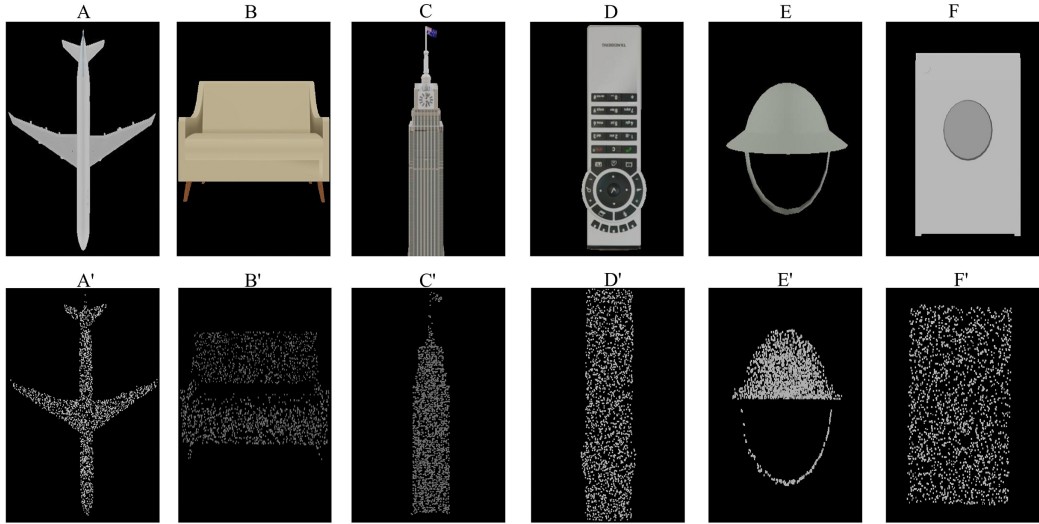

**Figure 3 Visualization of images and point clouds from our ShapeNet3D-CMA dataset.** (A′) to (F′) represent point clouds, while (A) to (F) are their corresponding rendered images.

$\lambda$ is a hyperparameter used to balance these two losses. In our experiments, we set $\lambda = 0.0001$.

## EXPERIMENTS

### Dataset construction

Existing publicly available point cloud and image datasets typically do not provide point cloud and image data in the same viewpoint, nor do they contain mapping relationships between points and image pixels. For this reason, we construct a cross-modal feature learning dataset, ShapeNet3D-Cross-modal Multi-view Alignment (ShapeNet3D-CMA), to support self-supervised learning of global semantic and local geometric features.

Our ShapeNet3D-CMA dataset is constructed based on the ShapeNet V2 dataset (*Chang et al., 2015*). We render the CAD models using Blender software by simulating virtual cameras to scan the models from multiple viewpoints around them. This process generates depth images and RGB-D images for each corresponding viewpoint. Subsequently, point clouds are reconstructed from the depth maps based on the rendering parameters. The constructed dataset includes rendered images from different viewpoints and the corresponding point clouds. Figure 3 shows some of the data in our ShapeNet3D-CMA dataset. Our dataset is publicly avaiable at https://zenodo.org/records/15269015.

We randomly select 41,928 samples from the rendered data to form the pre-training dataset. For each selected sample, the point cloud reconstructed with the maximum number of points from a specific viewpoint is chosen as the final sample.

### Downstream tasks

For downstream task evaluation, we validate our method's effectiveness against three distinct methods: (1) supervised methods, (2) masked reconstruction-based unsupervised

methods (*Liu et al., 2024*; *Xu et al., 2022*), *Wang et al. (2021)* targeting local geometric structures, and (3) unsupervised method (*Lin et al., 2024*) with joint global-local modeling capabilities that share similar optimization objectives with ours. All unsupervised methods follow standardized procedures: pre-training on our ShapeNet3D-CMA dataset, fine-tuning on downstream training sets, and reporting peak test accuracy—with the highest observed accuracy from multiple experimental trials recorded.

### Synthetic 3D object classification

ModelNet40 (*Yi et al., 2016*) is a meticulously annotated 3D CAD model dataset comprising 40 categories with a total of 12,311 models. The primary challenges of the ModelNet40 dataset stem from significant intra-class shape variations and diverse object poses and orientations. Derived from its synthetic data nature—characterized by absence of noise, well-defined shapes, and pristine backgrounds—this dataset emphasizes evaluating models' capability to capture precise local geometric and global semantic representations.

Following the standard protocol, we utilize 9,843 models for training and 2,468 models for testing. To ensure a fair comparison, our method uses only 1,024 points sampled from each model, containing solely coordinate information, as input. For reporting results, we employ the standard voting method (*Liu et al., 2019*) for final evaluation.

As shown in Table 1, our method achieves 93.2% accuracy, outperforming all comparable self-supervised methods, narrowing the gap with the state-of-the-art supervised method PointTransformer to merely 0.5%, while significantly surpassing classical supervised models such as PointNet++ and Dynamic Graph Convolutional Neural Network (DGCNN).

Experimental results demonstrate that our pre-training approach effectively learns precise local geometric and global semantic representations. Nevertheless, a performance gap persists when compared to top-tier supervised methods: Pyramid Vision Transformer (PVT) (93.6%) and PointTransformer (93.7%). We believe this is primarily because both PVT and PointTransformer utilize modified Transformer models, whereas our approach employs the standard Transformer model, and a performance gap still exists.

### Real-world 3D object classification

The ScanObjectNN dataset (*Qi et al., 2017a*) comprises 15 categories of objects scanned from real-world scenes. Its samples inherently present challenges such as object occlusions, background noise, and sensor artifacts. Consequently, models operating on this dataset must demonstrate the ability to accurately extract local geometric features amidst noise and occlusion disturbances, while also performing effective reasoning about the global semantics of complex scenes.

Following *Wang et al. (2021)*, we evaluate three variants: OBJ-BG preserves complete objects with injected background noise; OBJ-ONLY maintains full object clouds with artificial background removal; PB-T50-RS combines background interference with 50% random object occlusion while applying random spatial transformations.

**Table 1 Object classification results (%) on the ModelNet40 dataset.** Bold values indicate the best values.

| Supervised methods | Accuracy |
|---|---|
| PointNet (*Qi et al., 2017a*) | 89.2 |
| PointNet++ (*Qi et al., 2017b*) | 90.7 |
| DGCNN (*Wang et al., 2019*) | 92.9 |
| PointCNN (*Li et al., 2018*) | 92.5 |
| KPConv (*Thomas et al., 2019*) | 92.9 |
| RS-CNN (*Liu et al., 2019*) | 92.9 |
| PCT (*Guo et al., 2021*) | 93.2 |
| PVT (*Zhang et al., 2021*) | 93.6 |
| PointTransformer (*Engel, Belagiannis & Dietmayer, 2021*) | **93.7** |
| Transformer (*Yu et al., 2022*) | 91.4 |
| Transformer-OcCo (*Yu et al., 2022*) | 92.1 |
| Point-BERT (*Yu et al., 2022*) | 92.8 |
| Point-MAE (*Pang et al., 2022*) | 92.9 |
| Point-M2AE (*Zhang et al., 2022a*) | 92.7 |
| Inter-MAE (*Liu et al., 2024*) | 92.8 |
| **Ours** | **93.2** |

**Table 2 Object classification results (%) on the ScanObjectNN dataset.** Bold values indicate the best values.

| Methods | OBJ-BG | OBJ-ONLY | PB-T50-RS |
|---|---|---|---|
| PointNet (*Qi et al., 2017a*) | 73.3 | 79.2 | 68.0 |
| SpiderCNN (*Xu et al., 2018*) | 77.1 | 79.5 | 73.7 |
| PointNet++ (*Qi et al., 2017b*) | 82.3 | 84.3 | 77.9 |
| DGCNN (*Wang et al., 2019*) | 82.8 | 86.2 | 78.1 |
| PointCNN (*Li et al., 2018*) | 86.1 | 85.5 | 78.5 |
| GBNet (*Qiu, Anwar & Barnes, 2022*) | – | – | 80.5 |
| RepSurf-U (*Ran, Liu & Wang, 2022*) | – | – | 84.3 |
| Transformer (*Yu et al., 2022*) | 79.9 | 80.6 | 77.2 |
| Transformer-OcCo (*Yu et al., 2022*) | 84.9 | 85.5 | 78.8 |
| Point-BERT (*Yu et al., 2022*) | 87.4 | 87.1 | 82.4 |
| Point-MAE (*Pang et al., 2022*) | 88.1 | 87.9 | 84.0 |
| Point-M2AE (*Zhang et al., 2022a*) | 88.4 | 88.1 | **85.1** |
| Inter-MAE (*Liu et al., 2024*) | **89.3** | 87.9 | 83.3 |
| **Ours** | 88.5 | **88.7** | 84.5 |

Table 2 demonstrates that our method achieves state-of-the-art performance on the OBJ-ONLY variant, while attaining competitive results on OBJ-BG (0.8% lower than Inter-MAE) and PB-T50-RS (0.6% below Point-M2AE). These findings indicate that our approach effectively learns both local geometric and global semantic representations,

though direct comparisons with Inter-MAE and Point-M2AE reveal context-dependent strengths and limitations.

Compared to Inter-MAE: despite trailing by 0.8% on OBJ-BG, our method surpasses Inter-MAE by 1.2% on the most challenging PB-T50-RS variant. Crucially, the performance drop when switching from OBJ-BG to PB-T50-RS is 6.0% for Inter-MAE but only 4.0% for our method. This demonstrates that the learned local and global representations from our pre-training approach exhibit superior robustness against spatial transformations and object occlusions/partiality present in PB-T50-RS.

Compared to Point-M2AE: while our method demonstrates marginal improvements on the OBJ-BG (+0.1%) and OBJ-ONLY (+0.6%) variants, it lags behind by 0.6% on the most challenging PB-T50-RS variant. This performance gap primarily stems from Point-M2AE's hierarchical encoder architecture. Its multi-scale feature fusion mechanism excels at capturing local geometric details across scales, a capability critically important for handling complex scenarios characterized by object occlusion and partiality. In contrast, our approach relies on a standard Transformer encoder, which inherently struggles with extracting multi-scale local features essential for this difficult setting.

### Few-shot classification

To validate the generalization and transferability of the pretrained representations, we perform fine-tuning with extremely limited labeled samples, forcing the model to exclusively rely on the prior knowledge acquired during pre-training. Following the evaluation protocol in *Xu et al. (2022)*, we construct four few-shot learning tasks (covering different category-sample combinations) on the ModelNet40 dataset (*Wu et al., 2015*). For each task configuration, we conduct 10 independent experimental trials and report the mean classification accuracy along with its standard deviation.

As shown in Table 3, our method achieves higher average accuracy than all methods under the 5-way 10-shot, 10-way 10-shot, and 10-way 20-shot settings, while demonstrating the smallest variance among all self-supervised methods. In the 5-way 10-shot scenario, our average accuracy is second only to Point-MAE. These experimental results indicate that our pre-training approach learns more generic representations, delivering stronger performance in data-limited scenarios.

### Part segmentation

ShapeNetPart (*Cheng et al., 2023*) is the core benchmark dataset for 3D object part segmentation. Its primary task is to assign a part semantic label to each point within a point cloud data. This dataset encapsulates several key challenges: imbalanced part scales, cross-category semantic ambiguity (geometrically identical structures possessing different semantics across object categories), and blurred part boundaries. These challenges demand robust model capabilities encompassing multi-scale feature capture, context-aware global semantic understanding, and discrimination of local geometric features across diverse objects.

**Table 3 Few-shot classification results (%) on the ModelNet40 dataset.** Bold values indicate the best values.

| Methods | 5-way, 10-shot | 5-way, 20-shot | 10-way, 10-shot | 10-way, 20-shot |
|---|---|---|---|---|
| DGCNN-rand (*Wang et al., 2021*) | $31.6 \pm 2.8$ | $40.8 \pm 4.6$ | $19.9 \pm 2.1$ | $16.9 \pm 1.5$ |
| DGCNN-OcCo (*Wang et al., 2021*) | $90.6 \pm 2.8$ | $92.5 \pm 1.9$ | $82.9 \pm 1.3$ | $86.5 \pm 2.2$ |
| Transformer-rand (*Yu et al., 2022*) | $87.8 \pm 5.2$ | $93.3 \pm 4.3$ | $84.6 \pm 5.5$ | $89.4 \pm 6.3$ |
| Transformer-OcCo (*Yu et al., 2022*) | $94.0 \pm 3.6$ | $95.9 \pm 2.3$ | $89.4 \pm 5.1$ | $92.4 \pm 4.6$ |
| Point-BERT (*Yu et al., 2022*) | $94.6 \pm 3.1$ | $96.3 \pm 2.7$ | $91.0 \pm 5.4$ | $92.7 \pm 5.1$ |
| Point-MAE (*Pang et al., 2022*) | $\mathbf{96.2 \pm 2.5}$ | $97.4 \pm 1.6$ | $91.7 \pm 5.2$ | $94.6 \pm 3.5$ |
| Point-M2AE (*Zhang et al., 2022a*) | $88.1 \pm 5.9$ | $94.6 \pm 2.5$ | $85.0 \pm 6.5$ | $91.2 \pm 4.4$ |
| Iner-MAE (*Liu et al., 2024*) | $95.1 \pm 2.6$ | $97.5 \pm 1.7$ | $91.6 \pm 4.6$ | $94.1 \pm 3.7$ |
| **Ours** | $95.2 \pm 2.8$ | $\mathbf{98.1 \pm 1.4}$ | $\mathbf{91.8 \pm 4.5}$ | $\mathbf{94.8 \pm 3.5}$ |

In our experiments, following *Xue et al. (2023)*, we report three key metrics: Intersection over Union (IoU) per category $mIoU_{avg}$, mean IoU per instance $mIoU_{ins}$, and mean IoU per category.

As shown in Table 4, our method achieves comparable performance to Point-M2AE in $mIoU_{avg}$ while outperforming all other comparative methods. For the $mIoU_{ins}$, our approach ranks second to Point-M2AE, trailing by a marginal 0.3% gap. Experimental results demonstrate that our method delivers competitive performance in part segmentation tasks, though there remains explicit room for refinement compared to Point-M2AE.

Compared to Inter-MAE, our approach achieves higher scores on both $mIoU_{ins}$ and $mIoU_{avg}$. Further analysis of categories exhibiting IoU differences >1% reveals: for e-phone and bag classes, our approach achieves +6.0% and +2.7% advantages over Inter-MAE respectively, validating its superior capability in discriminating local geometric structures—particularly evident in resolving boundary ambiguity in non-rigid soft-body deformation (bag) and adapting to microscopic components (e-phone). However, a performance gap of 2.4% is observed in the rocket category, indicating Inter-MAE's enhanced contour modeling capability for large-scale continuous surfaces through its fusion of unmasked image information. Qualitative results (Fig. 4) corroborate that within regions annotated with red boxes, our method demonstrates significantly higher precision in parsing local geometric structures.

### 3D object detection

To further evaluate the applicability of our method in real-world scenarios, we conducted experiments on indoor 3D object detection tasks. Following the PiMAE (*Chen et al., 2023*) approach, we first performed pre-training on the SUN RGB-D (*Song, Lichtenberg & Xiao, 2015*) dataset and subsequently fine-tuned on the ScanNetV2 (*Dai et al., 2017*) dataset. When processing SUN RGB-D dataset scenes containing 20,000 points, we employ scaled-down PointNet (*Qi et al., 2017a*) to extract 2,048 key points.

We replace the point cloud encoder with a three-layer standard Transformer module from 3D End-to-end Transformer Detection (3DETR) (*Misra, Girdhar & Joulin, 2021*), where each layer features a hidden dimension of 256 and a 4-head multi-head attention

**Table 4 Part segmentation results on the ShapeNetPart dataset.** We report $mIoU_{avg}$ (%) and $mIoU_{ins}$ (%), as well as the IoU (%) for each categories. Bold values indicate the best values.

| Methods | PointNet | PointNet++ | DGCNN | Transformer | Transformer-OcCo | Point-BERT | Point-MAE | Point-M2AE | Inter-MAE | Ours |
|---|---|---|---|---|---|---|---|---|---|---|
| | Wang et al. (2019) | Qiu, Anwar & Barnes (2022) | Ran, Liu & Wang (2022) | Liu et al. (2024) | Liu et al. (2024) | Liu et al. (2024) | Xu et al. (2022) | Liu et al. (2019) | Lin et al. (2024) | |
| Aero | 83.4 | 82.4 | 84.0 | 82.9 | 83.3 | 84.3 | 84.9 | **85.1** | 84.7 | 84.7 |
| Bag | 78.7 | 79.0 | 83.4 | 85.4 | 85.2 | 84.8 | 83.2 | **86.5** | 82.4 | 85.1 |
| Cap | 82.5 | 87.7 | 86.7 | 87.7 | 88.3 | 88.0 | 88.8 | 89.6 | 88.9 | **89.0** |
| Car | 74.9 | 77.3 | 77.8 | 78.8 | 79.9 | 79.8 | 80.1 | **80.9** | 80.4 | 80.2 |
| Chair | 89.6 | 90.8 | 90.6 | 90.5 | 90.7 | 91.0 | **91.5** | **91.5** | 91.3 | 91.2 |
| e-phone | 73.0 | 71.8 | 74.7 | 80.8 | 74.1 | 81.7 | 74.0 | 77.2 | 74.6 | **80.6** |
| Guitar | 91.5 | 91.0 | 91.2 | 91.1 | 91.9 | 91.6 | **92.2** | 92.0 | 91.8 | 91.6 |
| Knife | 85.9 | 85.9 | 87.5 | 87.7 | 87.6 | **87.9** | 87.6 | 87.8 | 87.4 | 87.8 |
| Lamp | 80.8 | 83.7 | 82.8 | 85.3 | 84.7 | 85.2 | 85.8 | **86.0** | 85.9 | 85.3 |
| Laptop | 95.3 | 95.3 | 95.7 | 95.6 | 95.4 | 95.6 | **96.1** | 96.0 | 96.0 | 95.8 |
| Motor | 65.2 | 71.6 | 66.3 | 73.9 | 75.5 | 75.6 | 75.4 | **75.7** | 74.4 | 75.1 |
| Mug | 93.0 | 94.1 | 94.9 | 94.9 | 94.4 | 94.7 | 95.3 | 94.9 | **95.6** | 94.8 |
| Pistol | 81.2 | 81.3 | 81.1 | 83.5 | 84.1 | 84.3 | 84.7 | 84.9 | **85.2** | 84.3 |
| Rocket | 57.9 | 58.7 | 63.5 | 61.2 | 63.1 | 63.4 | 64.4 | 60.6 | **64.8** | 62.4 |
| s-board | 72.8 | 76.4 | 74.5 | 74.9 | 75.7 | 76.3 | **77.0** | 76.0 | 75.9 | 76.7 |
| Table | 80.6 | 82.6 | 82.6 | 80.6 | 80.8 | 81.5 | 81.4 | **82.3** | 81.4 | 82.2 |
| $mIoU_{ins}$ | 83.7 | 85.1 | 85.2 | 85.1 | 85.6 | 85.7 | 85.9 | **86.2** | 85.8 | 85.9 |
| $mIoU_{avg}$ | 80.3 | 82.0 | 82.3 | 83.4 | 84.1 | 84.1 | 83.9 | **84.2** | 83.8 | **84.2** |

mechanism. To maintain dimensional consistency, we reduce the number of attention heads in the decoder from six to four while keeping all other architectural components unchanged. During the fine-tuning phase, we strictly adhered to 3DETR's (*Misra, Girdhar & Joulin, 2021*) training protocol.

As shown in Table 5, our approach outperforms PiMAE by 0.7% in $AP_{25}$ and 2.3% in $AP_{50}$. This improvement, particularly the more significant gain (+2.3%) on the challenging $AP_{50}$ metric, clearly demonstrates the superiority of our pre-training strategy. By combining two self-supervised tasks, our approach effectively overcomes the issue of insufficient encoding of global semantic information in PiMAE due to its reliance on a single masked reconstruction task.

## Reconstruction results

To validate the point cloud reconstruction performance of our method, we conducted visual comparisons of the reconstruction effects on the ShapeNet test set (*Chang et al., 2015*) for Point-MAE, Inter-MAE, and our method. Figure 5 shows the reconstruction results of pre-trained models from each method under a 60% point cloud masking rate. Our method significantly outperforms other methods in terms of reconstruction quality, further validating its superiority.

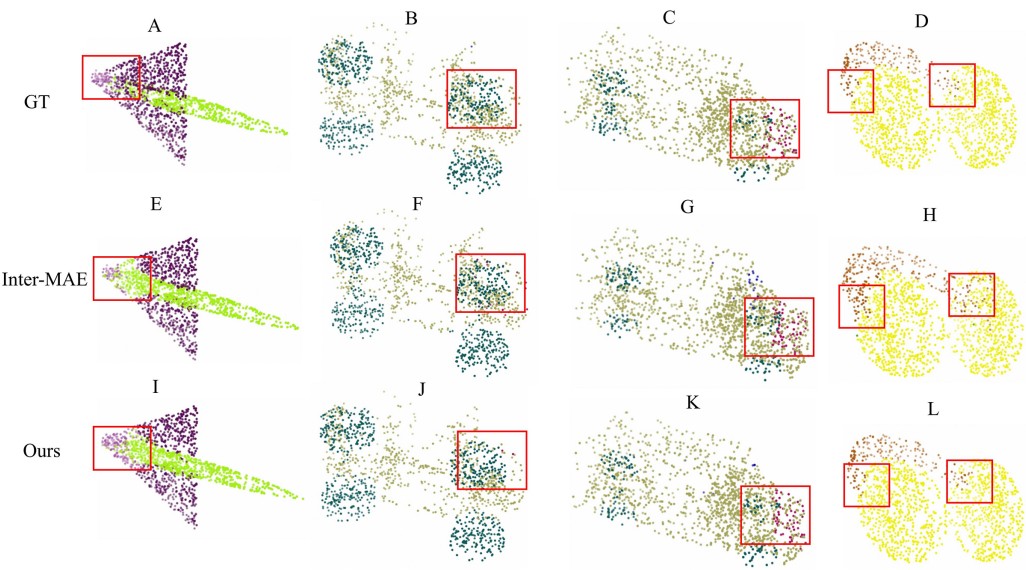

**Figure 4** **Qualitative results of part segmentation.** (A)–(D) represent the ground truth; (E)–(H) show the predictions from Inter-MAE; (I)–(L) demonstrate the results from our method.

**Table 5** **3D object detection performance on ScanNetV2 (*Dai et al., 2017*) val set.** We adopt the average precision(%) with 3DIoU thresholds of 0.25 ($AP_{25}$) and 0.5 ($AP_{25}$) for the evaluation metrics. Bold values indicate the best values.

| Methods | Pre-trained | ScanNetV2 | |
|---|---|---|---|
| | | $AP_{25}$ | $AP_{50}$ |
| DSS (*Song & Xiao, 2016*) | None | 15.2 | 6.8 |
| 3D-SIS (*Hou, Dai & Nießner, 2019*) | None | 40.2 | 22.5 |
| VoteNet (*Qi et al., 2019*) | None | 58.6 | 33.5 |
| 3DETR (*Misra, Girdhar & Joulin, 2021*) | None | 62.1 | 37.9 |
| PiMAE (*Chen et al., 2023*) | SUN RGB-D | 62.6 | 39.4 |
| +Ours | SUN RGB-D | **63.3 (+0.7)** | **41.7 (+2.3)** |

## Ablation study

All ablation studies were conducted on the object classification task using the most challenging PB-T50-RS variant of the ScanObjectNN dataset (*Qi et al., 2017a*). The highest classification accuracy observed over multiple experimental runs is reported.

### Percentage of masks

To investigate the impact of masking ratios on classification accuracy,we conducted experiments with different masking proportions. As indicated in Table 6, our method achieves optimal performance at a 60% masking ratio with 84.5% classification accuracy. Experimental results demonstrate that both excessively high and low masking ratios lead to performance degradation.

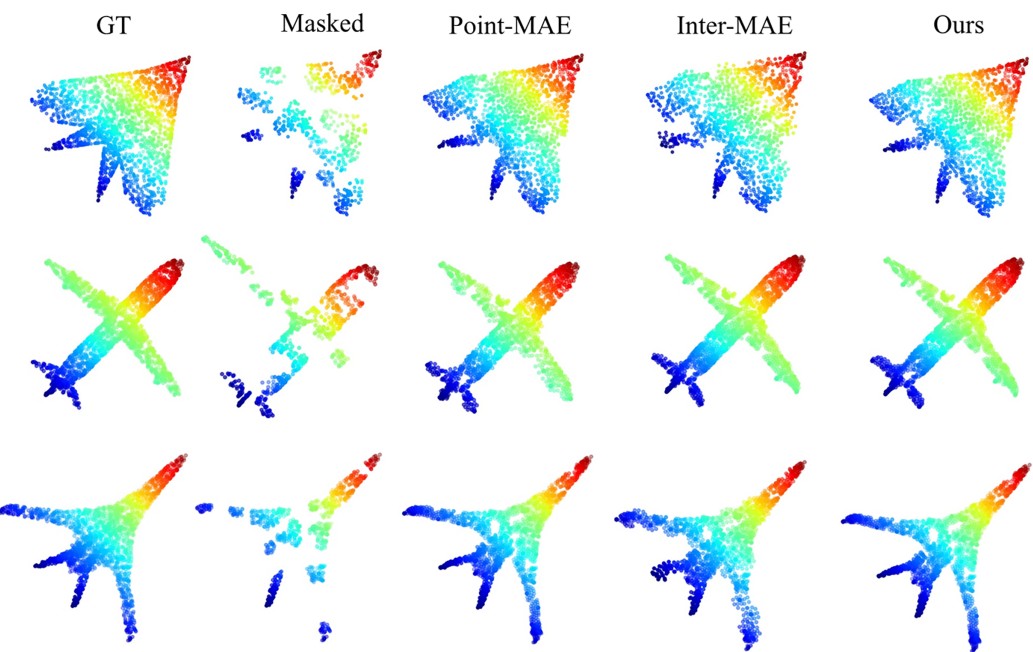

GT | Masked | Point-MAE | Inter-MAE | Ours

**Figure 5** **Reconstruction results on the ShapeNet test set.**

**Table 6** **Classification accuracy with different mask ratios.**

| Ratio | Accuracy (%) |
| --- | --- |
| 40% | 83.8 |
| 60% | 84.5 |
| 80% | 84.2 |
| 90% | 84.3 |

Excessively high masking ratios critically impair point cloud understanding by obliterating essential local geometric structures and disrupting object continuity. This dual degradation prevents models from effectively reasoning about global semantics and reconstructing masked regions. In our method, high masking ratios applied to point clouds necessitate proportionally aggressive masking of images. This inevitably compromises global semantic extraction from the visual modality. Under such conditions, enforcing cross-modal semantic alignment tasks introduces disruptive noise rather than meaningful learning signals.

Conversely, excessively low masking ratios create information overload, fostering lazy learning behaviors where models exploit superficial patterns instead of learning robust representations.

### Masking strategy

In this experiment, a fixed random masking ratio of 60% was applied to the point cloud, while three distinct masking strategies were compared for the image modality: (1) random masking; (2) correspondence masking (the masked image regions correspond to the

**Table 7 Classification accuracy using different masking strategies.**

| Point cloud | Image | Accuracy (%) |
|---|---|---|
| Random | Random | 84.1 |
| Random | Correspondence | 84.2 |
| Random | Complementary | 84.5 |

masked point cloud regions); (3) complementary masking (the masked image regions correspond to the visible point cloud regions).

As shown in Table 7, the complementary masking strategy achieved the best performance with an accuracy of 84.5%, outperforming corresponding masking by +0.3% and random masking by +0.4%. The superiority of complementary masking primarily stems from its synergistic enhancement of the two self-supervised tasks:

(1) Optimization of Local Mask Reconstruction: when local geometric information in point clouds is masked, complementary masking retains 2D features (*e.g.*, textures, edges) in the image corresponding to the masked point cloud regions. This provides critical auxiliary information for the point cloud reconstruction task.

(2) Enhancement of Global Semantic Alignment: under complementary masking, the globally encoded semantic information in the image forms a correspondence with the semantic information of the masked regions in the point cloud. This design forces the model to infer the global semantics of masked regions based solely on visible areas of the point cloud, thereby reinforcing its representational capacity for global semantic features.

### Selection of self-supervised tasks

To evaluate the impact of global semantic alignment and local mask reconstruction as self-supervised tasks on representation learning, we compared different task combinations and varying $\lambda$ values on classification accuracy by adjusting the weighting coefficient in Eq. (10). During pre-training, we observed a significant magnitude discrepancy between the local mask reconstruction loss ($Loss_{pc}$) and the global semantic alignment loss ($Loss_f$), with $Loss_f$ being approximately 1,000 times larger than $Loss_{pc}$. Therefore, setting $\lambda = 0.001$ ensured comparable contributions from $loss_{pc}$ and $\lambda * loss_f$ to the total loss magnitude. Experimental results are summarized in Table 8.

Experimental results demonstrate that models using only local mask reconstruction task significantly outperform those relying solely on global semantic alignment task, while combining both tasks yields clear advantages over either single-task configuration. This not only confirms the critical importance of locally learned geometric features for classification accuracy but also reveals complementary benefits in representation learning between the two tasks.

When $\lambda = 0.001$, the contributions of the two losses to the total loss reach equilibrium; when $\lambda$ decreases to 0.0001, the contribution of the global semantic alignment task reduces by a factor of ten, yet the classification accuracy increases by 0.3 percentage points. This suggests an intrinsic primary-secondary relationship between the two tasks (with local

**Table 8 Classification accuracy for different self-supervised tasks.**

| $Loss_{pc}$ | $Loss_f$ | $\lambda$ | Accuracy (%) |
|---|---|---|---|
| – | ✓ | – | 81.7 |
| ✓ | – | – | 84.2 |
| ✓ | ✓ | 0.001 | 84.3 |
| ✓ | ✓ | 0.0001 | 84.5 |

**Table 9 comparison of parameters and training time of self-supervised-based methods (pre-training + fine-tuning).** Bold values indicate the best values.

| Method | Params (M) | Time (s) |
|---|---|---|
| Point-BERT (*Liu et al., 2024*) | 69.02 + 22.06 | 0.233 + **0.019** |
| Point-MAE (*Xu et al., 2022*) | 29.00 + 22.09 | **0.220** + 0.021 |
| Point-M2AE (*Wang et al., 2021*) | **15.25** + **12.83** | 0.383 + 0.044 |
| Inter-MAE (*Lin et al., 2024*) | 35.07 + 22.22 | 0.780 + 0.022 |
| Ours | 57.69 + 22.09 | 0.381 + 0.032 |

mask reconstruction as primary and global semantic alignment as secondary). By selecting an appropriate $\lambda$ to regulate the contributions of both tasks, the model's representation learning capability can be effectively enhanced.

## Efficiency comparison across methods

To evaluate the efficiency of the pre-training and fine-tuning stages, we measured the parameter count for both the pre-trained and fine-tuned models of our method and other self-supervised approaches, along with the time required for a single forward pass. Evaluation during the fine-tuning stage was uniformly conducted on the fine-tuned ModelNet40 classification task model. Batch size for pre-training and fine-tuning stages were set to 64 and 32, respectively. All timing metrics were measured and reported on a single NVIDIA RTX A6000 GPU.

As shown in Table 9, Point-M2AE achieves the smallest parameter count in both pre-training and fine-tuning stages. Regarding computational efficiency, Point-MAE and Point-BERT exhibit the shortest forward pass time during pre-training and fine-tuning phases, respectively.

Our method requires a dedicated image encoder to process visual data, along with a dual-branch decoder that separately extracts global semantic features and local geometric features from point clouds. This architectural design results in higher model complexity and a larger parameter count during pre-training.

It is noteworthy that although Inter-MAE also employs an additional image encoder, it lacks a dedicated global semantic decoder, resulting in lower parameter count than our method. In execution time profiling, we observe that during a single pre-training forward pass, Inter-MAE consumes approximately half its processing time on image encoding. By contrast, our approach applies masking to images as well, significantly reducing image

encoding overhead. Consequently, our method demonstrates notable advantages in pre-training efficiency compared to Inter-MAE.

## CONCLUSION

This study proposes a cross-modal self-supervised point cloud representation learning framework that integrates global semantic alignment with local masked reconstruction. This combined approach effectively facilitates the joint optimization of global semantic understanding and local geometric feature learning within point clouds. Experimental results demonstrate that the proposed method achieves outstanding performance across multiple downstream tasks, including 3D object classification, few-shot learning, and part segmentation. Furthermore, the multimodal dataset constructed in this study provides a new benchmark resource for cross-modal research.

Our method's pre-training on synthetic CAD data inevitably induces domain shift, as evidenced by significant accuracy drops when transferring object classification from ModelNet40 to ScanObjectNN. Enhancing real-world robustness is thus critical. Furthermore, our encoder adopts a standard Transformer architecture, which exhibits limitations in multi-scale feature encoding compared to customized variants. Future work should explore heterogeneous encoder structures to overcome the constraint of single-scale feature representation.

### Funding

This research was funded by the Fundamental Research Funds for the Central Universities (No. 3132025278), the Shandong Natural Science Foundation of China (No. ZR2024QF255), and the Yantai Science and Technology Innovation Development Plan (No. 2024YT06000226). The funders had no role in study design, data collection and analysis, decision to publish, or preparation of the manuscript.

### Grant Disclosures

The following grant information was disclosed by the authors:
Fundamental Research Funds for the Central Universities: 3132025278.
Shandong Natural Science Foundation of China: ZR2024QF255.
Yantai Science and Technology Innovation Development Plan: 2024YT06000226.

### Competing Interests

The authors declare that they have no competing interests.

### Author Contributions

- Fei Wang conceived and designed the experiments, performed the experiments, analyzed the data, prepared figures and/or tables, authored or reviewed drafts of the article, and approved the final draft.

- Xingzhen Dong performed the experiments, performed the computation work, prepared figures and/or tables, authored or reviewed drafts of the article, and approved the final draft.
- Jia Wu conceived and designed the experiments, performed the experiments, analyzed the data, performed the computation work, prepared figures and/or tables, and approved the final draft.
- Weishi Zhang conceived and designed the experiments, analyzed the data, authored or reviewed drafts of the article, and approved the final draft.
- Tuo Zhou conceived and designed the experiments, performed the computation work, authored or reviewed drafts of the article, and approved the final draft.

## Data Availability

The data is available at Zenodo: dong,. xingzhen. (2025). ShapeNet3D-CMA [Data set]. Zenodo. https://doi.org/10.5281/zenodo.15269015.

The code files are available in the Supplemental File.

## Supplemental Information

Supplemental information for this article can be found online at http://dx.doi.org/10.7717/peerj-cs.3194#supplemental-information.

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
