# Peer review of "CrossAlignNet: a self-supervised feature learning framework for 3D point cloud understanding"

_PeerJ Computer Science, doi:10.7717/peerj-cs.3194_

## Round 0.1 · original submission · Major Revisions

· Academic Editor

Major Revisions

Dear authors,

You are advised to critically respond to all comments point by point when preparing an updated version of the manuscript and while preparing for the rebuttal letter. Please address all comments/suggestions provided by reviewers, considering that these should be added to the new version of the manuscript.

Kind regards,
PCoelho

**Language Note:** The review process has identified that the English language must be improved. PeerJ can provide language editing services - please contact us at [email protected] for pricing (be sure to provide your manuscript number and title). Alternatively, you should make your own arrangements to improve the language quality and provide details in your response letter. – PeerJ Staff

Reviewer 1 ·

Basic reporting

1. Please clarify the main difference between your method and PiMAE/Inter-MAE.
2. The limitations and future work should be added to the revision.
3. The pre-training and fine-tuning parameters and times should also be compared.
4. Several surveys should be added to the revised paper:
a. Zeng, Changyu, et al. "Self-supervised learning for point cloud data: A survey." Expert Systems with Applications 237 (2024): 121354.
b. Xiao, Aoran, et al. "Unsupervised point cloud representation learning with deep neural networks: A survey." IEEE Transactions on Pattern Analysis and Machine Intelligence 45.9 (2023): 11321-11339.

Experimental design

1. Classification experiments on ModelNet40 should also be conducted.
2. Point-M2AE should also be added to the comparison.

Validity of the findings

no comment

Reviewer 2 ·

Basic reporting

The article has certain strengths, but the following questions arise:



9. Other minor issues; There is a spelling error on line 343; It is unclear why the references do not have hyperlinks; this might be a template issue; The equal signs in Equations 7 and 8 are not aligned.

Experimental design

3. The use of the joint loss function; Is the joint loss function only used in self-supervised tasks, or is it applied in all tasks? Additionally, regarding the experiments on the hyperparameter λ, the results of 0 × Loss_f + 1 × Loss_rec seem to be similar to those of 1 × Loss_f + 0.001 × Loss_rec, which is a confusing conclusion. More parameters should be included, such as from 0.9 × Loss_f + 0.1 × Loss_rec to the experimental settings in the manuscript. Otherwise, the manuscript should explain why such a small hyperparameter was chosen.

5. The analysis of experimental results; The manuscript demonstrates the superiority of the proposed method in multiple tasks and datasets, which is good. However, the analysis of the experimental results is relatively simple. Could specific analyses of the method's superiority be added, starting from the proposed method and comparing it with other methods? For example, in the object classification task, it is mentioned that "It shows that the proposed method is more advantageous when dealing with complex geometric transformations (e.g., rotation, scaling) and partially occluded scenes." What is the reason behind this statement? Which data are related to complex geometric transformations (since only overall results are provided without analysis of simple transformations and complex geometric transformations)? Therefore, every conclusion in the manuscript should be supported by experiments or methodological explanations, rather than just a summary.

6.The design of Table 3; The design of Table 3 is unreasonable. The table headers should correspond to the data. For example:
aero lamp bag laptop ……
83.4 80.8 78.7 95.3 ……
```

7. Regarding statistical analysis; In several important experiments, the advantages in Table 3 and Table 2 are very small compared to the referenced methods. Statistical analysis should be introduced to demonstrate that these advantages are not random.

Validity of the findings

1. It is suggested that experiments should be conducted to demonstrate the superiority of the proposed framework and method by addressing the complementary masks, namely, the ablation experiments on the masks of point clouds and images.

2. Regarding the selection of 3D and 2D masks; And how to ensure that they are complementary, can this be illustrated with figures? For readers, it requires a strong imagination to understand solely based on textual descriptions.


4. The new dataset; The manuscript mentions the construction of a new 3D point cloud dataset, but it has not been tested in the experiments. Why is that? The necessity of this dataset should not only be explained through descriptions and figures but also proven through experiments. This is the biggest issue of this paper.


8. About he conclusion section; It is suggested that the conclusion section should discuss the limitations of the proposed method and future work directions.

---

## Round 0.2 · Minor Revisions

· Academic Editor

Minor Revisions

Dear authors,

Thanks a lot for your efforts to improve the manuscript.

Nevertheless, some concerns remain that need to be addressed.

Like before, you are advised to critically respond to the remaining comments point by point when preparing a new version of the manuscript and while preparing for the rebuttal letter.

Kind regards,
PCoelho

Reviewer 1 ·

Basic reporting

1. The paper is well-written.
2. The references are sufficient.

Experimental design

The experiments are comprehensive. However, could your method be generalized to indoor scene tasks, such as 3D semantic segmentation and 3D object detection? MaskPoint and Point-M2AE conduct these two tasks, respectively.

Validity of the findings

The findings are validated in the submission.

Additional comments

Thanks for the rebuttal; most of my concerns have been answered. I would like to see more results on indoor scene tasks, including ScanNet and S3DIS.

---

## Round 0.3 · accepted · Accept

· Academic Editor

Accept

Dear authors, we are pleased to verify that you meet the reviewer's valuable feedback to improve your research.

Thank you for considering PeerJ Computer Science and submitting your work.

The references suggested by R1 are not required.

Kind regards
PCoelho

Reviewer 1 ·

Basic reporting

The several point cloud SSL methods should be discussed in the final version:
1. Fei, B., Yang, W., Liu, L., Luo, T., Zhang, R., Li, Y., & He, Y. (2023). Self-supervised learning for pre-training 3d point clouds: A survey. arXiv preprint arXiv:2305.04691.
2. Fei, B., Luo, T., Yang, W., Liu, L., Zhang, R., & He, Y. (2024). Curriculumformer: Taming Curriculum Pre-Training for Enhanced 3-D Point Cloud Understanding. IEEE Transactions on Neural Networks and Learning Systems, 36(4), 7316-7330.
3. Zhang, Q., & Hou, J. (2023). Pointvst: Self-supervised pre-training for 3d point clouds via view-specific point-to-image translation. IEEE Transactions on Visualization and Computer Graphics, 30(10), 6900-6912.
4. Zhang, Y., Hou, J., Ren, S., Wu, J., Yuan, Y., & Shi, G. (2025). Self-supervised learning of lidar 3d point clouds via 2d-3d neural calibration. IEEE Transactions on Pattern Analysis and Machine Intelligence.

Experimental design

no comment

Validity of the findings

no comment